# Transient external force induces phenotypic reversion of malignant epithelial structures via nitric oxide signaling

Benjamin L Ricca[1†], Gautham Venugopalan[1†], Saori Furuta[2], Kandice Tanner[2,3‡], Walter A Orellana[2,3], Clay D Reber[1], Douglas G Brownfield[1,2§], Mina J Bissell[2*], Daniel A Fletcher[1,2*]

[1]Bioengineering Department and Biophysics Program, University of California, Berkeley, Berkeley, United States; [2]Biological Systems and Engineering Division, Lawrence Berkeley National Laboratory, Berkeley, United States; [3]Center for Cancer Research, National Cancer Institute, National Institutes of Health, Bethesda, United States

**\*For correspondence:**
mjbissell@lbl.gov (MJB);
fletch@berkeley.edu (DAF)

[†]These authors contributed equally to this work

**Present address:** [‡]Laboratory of Cell Biology, Center for Cancer Research, National Cancer Institute, Bethesda, United States; [§]Department of Biochemistry, School of Medicine, Stanford University, Stanford, United States

**Competing interests:** The authors declare that no competing interests exist.

**Abstract** Non-malignant breast epithelial cells cultured in three-dimensional laminin-rich extracellular matrix (lrECM) form well organized, growth-arrested acini, whereas malignant cells form continuously growing disorganized structures. While the mechanical properties of the microenvironment have been shown to contribute to formation of tissue-specific architecture, how transient external force influences this behavior remains largely unexplored. Here, we show that brief transient compression applied to single malignant breast cells in lrECM stimulated them to form acinar-like structures, a phenomenon we term 'mechanical reversion.' This is analogous to previously described phenotypic 'reversion' using biochemical inhibitors of oncogenic pathways. Compression stimulated nitric oxide production by malignant cells. Inhibition of nitric oxide production blocked mechanical reversion. Compression also restored coherent rotation in malignant cells, a behavior that is essential for acinus formation. We propose that external forces applied to single malignant cells restore cell-lrECM engagement and signaling lost in malignancy, allowing them to reestablish normal-like tissue architecture.

DOI: https://doi.org/10.7554/eLife.26161.001

## Introduction

Acinar morphogenesis involves integration of both biochemical and biophysical cues, the disruption of which can lead to malignancy (*Bello-DeOcampo et al., 2001*; *Bissell and Hines, 2011*). Acinar formation can be modeled in three dimensions by embedding mammary epithelial cells in laminin-rich extracellular matrix (lrECM) gels (*Barcellos-Hoff et al., 1989*). Under these conditions, primary or non-malignant cells form polarized, growth-arrested acini after 7 to 10 days in culture, whereas breast cancer cells grow continuously to form large disorganized colonies (*Petersen et al., 1992*). This developmental process can be influenced by the passive mechanical properties of the culture microenvironment. Normal or non-malignant breast epithelial cells grown on very stiff substrata form phenotypically malignant, non-polarized structures (*Chaudhuri et al., 2014*; *Paszek et al., 2005*). The importance of the mechanics of the ECM microenvironment can be seen dramatically when cells are switched from stiff pure collagen gels where they cannot produce milk to soft substrata, including lrECM, where milk proteins can be expressed (*Alcaraz et al., 2008*).

However, active mechanical inputs such as mechanical compression can elicit responses distinct from passive mechanical properties. Compression has been shown to serve as an external signal that regulates the structure and behavior of other tissue types. Step compression of whole cartilage tissue increases interleukin expression (*Murata et al., 2003*), whereas dynamic compression of bone enhances bone remodeling (*Chamay and Tschantz, 1972*; *Wolff, 1892*). How such active mechanical inputs influence acinar morphogenesis is yet to be explored, and we hypothesized that active compression could alter the growth and development of breast epithelium.

To explore this hypothesis, we used the HMT3522 progression series developed by Briand and colleagues (*Briand et al., 1987*; *Briand et al., 1996*). We applied a short-timescale compression to single malignant T4-2 cells and non-malignant S1 cells embedded in lrECM. Our cell culture system provides a model for malignant vs. non-malignant behavior that has been widely used, although it does not encompass the full spectrum of cellular behaviors observed in malignancy in vivo. Following compression, cells were cultured for up to 10 days. Surprisingly, we found that this transient compression can phenotypically revert malignant cells, a phenomenon we term 'mechanical reversion.' We also show that this mechanically induced shift in phenotype is mediated by nitric oxide signaling. Our findings suggest that well-timed mechanical stimuli may have some beneficial effect in the inhibition of cancer progression.

## Results

### Transient compression phenotypically reverts malignant breast epithelial cells

To directly test the effects of externally applied forces on acinar morphogenesis, we grew epithelial cells in deformable silicone wells to apply defined strains. We embedded single non-malignant S1 or malignant T4-2 breast epithelial cells in lrECM gels, as described previously (*Lee et al., 2007*), and polymerized the lrECM gel in pre-stretched silicone wells with open tops covered in media (*Figure 1—figure supplement 1F*) (*Brownfield et al., 2013*). Thirty minutes after polymerization, the pre-stretch was removed to apply a compressive strain to the lrECM and embedded cells (*Figure 1A*). Rheological measurements showed that the stress generated by the compressive strain relaxed within minutes (*Figure 1—figure supplement 1G*).

We grew non-malignant S1 and malignant T4-2 cells in our deformable silicone wells without compression (*Figure 1B–i,ii*) and observed that uncompressed T4-2 colonies were disorganized and significantly larger than uncompressed S1 colonies that formed polarized acini (*Figure 1C*; analysis of variance, $p=1.6\times10^{-5}$), consistent with previous data (*Petersen et al., 1992*; *Weaver et al., 1997*). We then applied the step compression as described above to T4-2 cells shortly after embedding them in the lrECM and compared their growth to that of the uncompressed S1 and T4-2 colonies (*Figure 1B–i–iii*). Surprisingly, compression of T4-2 cells at the single-cell stage led to a significant reduction in colony size 10 days later (*Figure 1C* and *Figure 1—figure supplement 2I*). Many of the colonies were not detectably different in size from non-malignant S1 acini. In contrast, compressed S1 acini were not significantly different from uncompressed S1 colonies.

Non-malignant cells exit the cell cycle at late stages of acinar morphogenesis as measured by the absence of Ki67, a protein found in all stages of cell cycle (*Lelièvre et al., 1998*), whereas malignant cells continue to express Ki67 since they do not undergo growth-arrest. To test whether compression simply slowed proliferation or drove the cells to exit the cell cycle, we fixed cells after the compression assay and stained them for Ki67. Consistent with phenotypic reversion, compressed T4-2 colonies formed growth-arrested structures with a lower percentage of Ki67-positive cells than uncompressed T4-2 cells (*Figure 1D*; t-test, p=0.012).

We performed a qualitative counting analysis of colony morphology after 10 days of culture (*Figure 1E*) aiming to demonstrate the heterogeneity within the populations. With compression, the majority of T4-2 cells grew into smaller, more organized colonies which were similar to those treated with an inhibitor for epidermal growth factor receptor (Tyrphostin, AG 1478) or PI3K (LY294002) (*Figure 1B–iii* and *Figure 1—figure supplement 3*).

Breast epithelial cells cultured in the '3D-on-top' geometry have been shown to behave similarly to those grown in the 3D embedded geometry by previous studies (*Lee et al., 2007*; *Liu et al., 2004*). After attachment to the pre-polymerized lrECM gel, cells in the '3D-on-top' geometry are

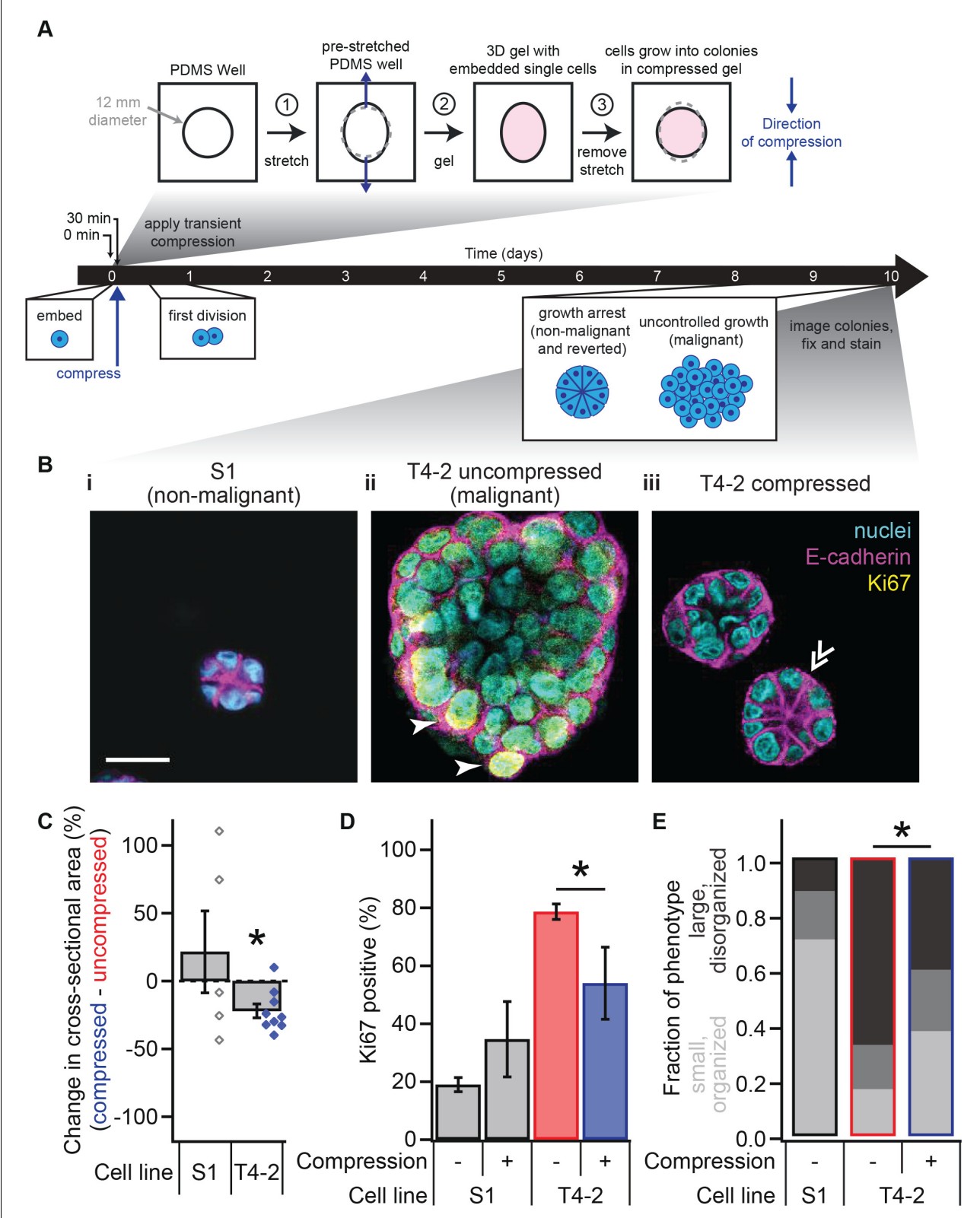

**Figure 1.** Transient compression of breast epithelial cells grown in laminin-rich ECM reverts the malignant phenotype. (**A**) Cells were compressed after 30 min of lrECM polymerization at 37°C, before first division (day 1) or formation of growth-arrested colonies or large disorganized colonies (day 10). Compressive strain dissipated on a minutes time scale (*Figure 1—figure supplement 1*) and affected cells grown in a modified culture geometry (*Figure 1—figure supplement 2*). (**B**) Representative confocal immunofluorescence images of uncompressed S1 (**i**) and T4-2 colonies (**ii**), and

*Figure 1 continued on next page*

Figure 1 continued

compressed T4-2 colonies (iii). Solid arrowheads indicate cells that stained positive for Ki67. Double arrowheads indicate a T4-2 colony with its E-cadherin arranged in a 'star pattern' through its central section. Scale bar 25 µm. Compressed T4-2 colonies resemble T4-2 cells treated with chemical reverting agents (*Figure 1—figure supplement 3*). (C) 23% compression of single malignant cells led to reduction in colony size (N = 5, 9 pairs of gels, bars are mean ± SEM, points are means of individual experiments, with 90 ± 44 colonies per sample [mean ± SD]). *Difference is significant at p<0.05 in paired t-test, comparing compressed and uncompressed samples in same chamber preparation (see also *Figure 1—figure supplement 2*). Analogous experiments using MCF10A cancer progression series cells gave similar results (*Figure 1—figure supplement 4*). (D) Colonies grown from compressed single cells exhibited enhanced growth arrest (N = 4, 4, 14, and 6 gels, mean ± SEM, with 33 ± 12 colonies per sample [mean ± SD]), determined by the absence of Ki67 stain. *Difference is statistically significant at p<0.02 in two-sample t-test. (E) Bar graphs displaying qualitative analysis of phenotype of uncompressed S1, uncompressed T4-2, and compressed T4-2 colonies after 10 days of embedded culture in lrECM. Confocal immunofluorescence images were scored in a blinded fashion. Colonies were categorized as having a 'small, organized' phenotype (characteristic of normal development or reversion) if they contained at least three of the following four features: (1) a small size, (2) a round shape, (3) the absence of Ki67, and (4) well-organized E-cadherin cell-cell junctions (appearing as a star pattern in cross section) or a cleared lumen. Colonies with one or none of these four features were categorized as 'large, disorganized' (characteristic of malignant behavior). The increase in the proportion of colonies with an organized phenotype and the decrease in proportion of a disorganized phenotype in demonstrate that a fraction of the population of malignant T4-2 cells were phenotypically reverted with compression. This difference is statistically significant (*) between the T4-2 uncompressed and compressed (p<0.05, two-sample t-test, N = 3, 3 experiments). Each bar represents the average fraction of each phenotype category across 2, 3, and 3 gels, respectively, with 34 ± 13 colonies per gel (mean ± SD).
DOI: https://doi.org/10.7554/eLife.26161.002

The following source data and figure supplements are available for figure 1:

**Source data 1.** Colony size and proliferation status in compressed and uncompressed T4-2 and S1 cells.
DOI: https://doi.org/10.7554/eLife.26161.007
**Figure supplement 1.** Application of transient compression to 3D cultured breast epithelial cells.
DOI: https://doi.org/10.7554/eLife.26161.003
**Figure supplement 2.** Mechanical reversion by transient compression occurs in both 3D embedded and 3D-on-top culture geometries.
DOI: https://doi.org/10.7554/eLife.26161.004
**Figure supplement 3.** Chemical agents revert the malignant phenotype.
DOI: https://doi.org/10.7554/eLife.26161.005
**Figure supplement 4.** Transient compression reduces growth of in cultured malignant cells of the MCF10A series.
DOI: https://doi.org/10.7554/eLife.26161.006

coated with a thin layer of lrECM drip, creating a 3D-like microenvironment that is more amenable to imaging, as cells and colonies exist primarily within a single plane that is approximately parallel to the bottom of the chamber. We found that compression in the 3D-on-top geometry reduced the growth of malignant cells in a manner similar to that in the 3D embedded geometry (*Figure 1—figure supplement 2*; *Figure 1—figure supplement 2J*) and confirmed the statistical significance (Figure 3F and *Figure 3—figure supplement 1*; paired t-test p=0.01).

To determine whether this phenotypic shift owing to transient compression was specific to the cell line used, we tested the occurrence of mechanical reversion in a second cell line, the MCF10A breast cancer progression series (*Santner et al., 2001*). Non-malignant MCF10A cells formed polarized, growth-arrested acini in 3D on-top-lrECM culture, similar to S1 cells, whereas malignant MCF10A-CA1d cells (CA1d) grew into larger, disorganized colonies cells (*Debnath et al., 2003*). After compression, malignant CA1d cells showed a reduction in colony size, whereas non-malignant MCF10A cells did not (*Figure 1—figure supplement 4*; paired t-test p=0.007 and p=0.29, respectively). This result indicates that compression-induced phenotypic reversion is not unique to the HMT3522 series.

## Mechanical reversion occurs above a threshold strain

To investigate whether compression-induced changes to growth and colony size were dependent on the extent of compression, we varied the amount of step displacement on the deformable wells to test the effect of differential compressive strains ranging from 10–23% (*Figure 2A*). Compression resulted in a threshold response: At and above 15% of compressive strain, colonies decreased in size by about 25% (*Figure 2B*; paired t-test, p=$2.85 \times 10^{-4}$) and had significantly fewer cells (*Figure 2C*; paired t-test, p=0.012).

Since stiffness of the ECM can alter a multicellular phenotype significantly (*Chaudhuri et al., 2014*; *Paszek et al., 2005*), we measured the ECM stiffness by means of storage (elasticity) and loss

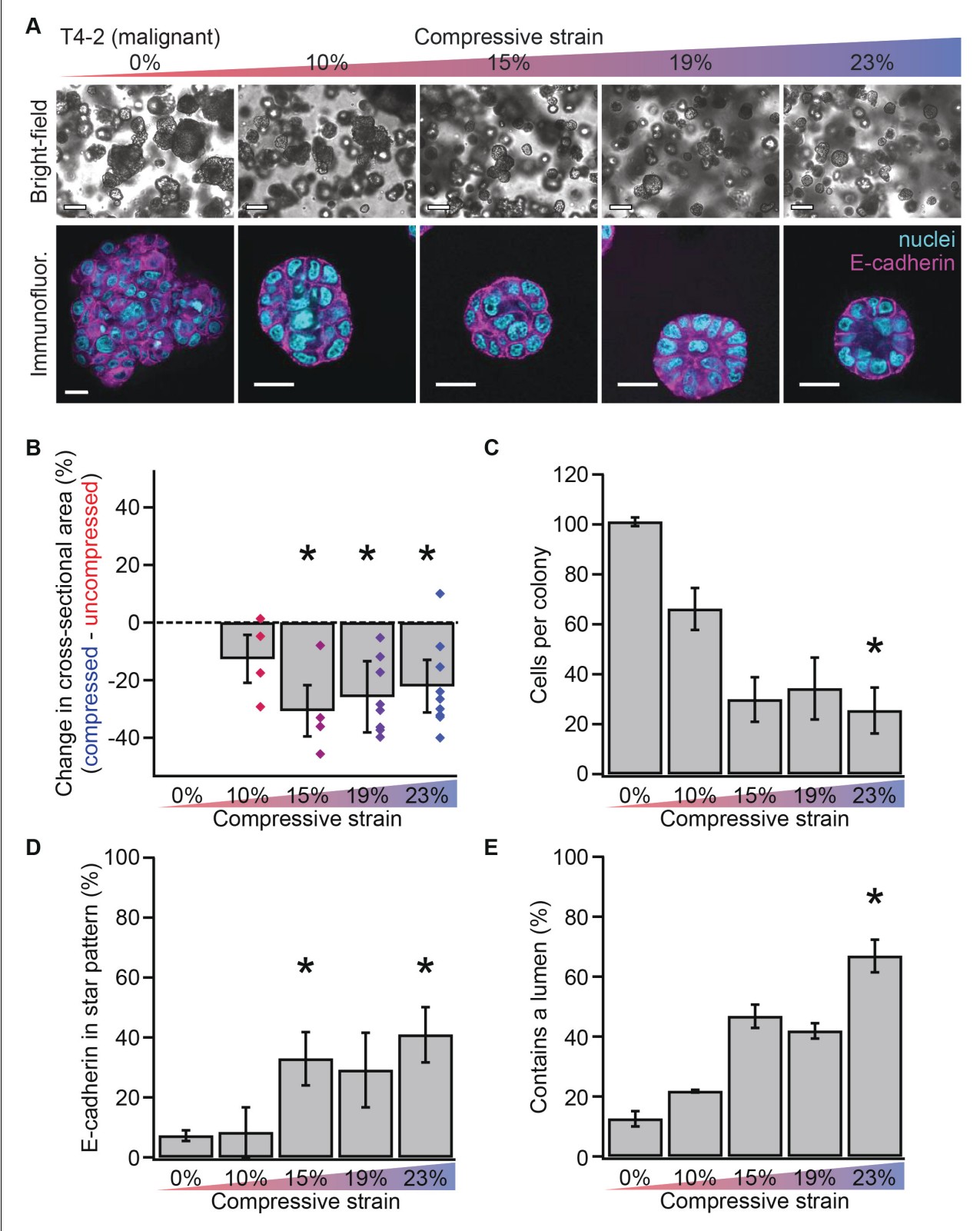

**Figure 2.** Mechanical reversion occurs above a threshold strain. (**A**) Bright-field and confocal immunofluorescence images of malignant T4-2 cell colonies grown for 10 days in compressed gels. Scale bars 100 μm bright-field and 25 μm confocal. (**B**) Colonies grown in compressed matrices were smaller (N = 4, 4, 7, and 9 gels, each with paired uncompressed control, bars mean ± SEM, points are means of individual experiments, with 121 ± 62 colonies per sample [mean ± SD]). *Differences are statistically significant at p<0.05 compared to control samples in paired t-test with a gel made from
*Figure 2 continued on next page*

*Figure 2 continued*
the same preparation. (C) Compressed colonies contained fewer cells as measured by counting nuclei (N = 11, 3, 3, 2, and 4 gels, mean ± SEM, with 40 ± 18 colonies per sample [mean ± SD]). *Difference is statistically significant at p<0.02 compared to control samples in paired t-test with a gel made from the same preparation. (D) Additionally, colonies grown in compressed matrices had multicellular architectures resembling non-malignant colonies (N = 15, 3, 3, 2, and 6 gels, mean ± SEM, with 39 ± 17 colonies per sample [mean ± SD]). *Differences are statistically significant at p<0.001 in analysis of variance. (E) Compression increased the frequency of lumen formation (N = 11, 3, 3, 2, and 4 gels, mean ± SEM, with 40 ± 18 colonies per sample [mean ± SD]). *Difference is statistically significant at p<0.001 in analysis of variance.
DOI: https://doi.org/10.7554/eLife.26161.008
The following source data is available for figure 2:

**Source data 1.** T4-2 colony size and architecture by compressive strain.
DOI: https://doi.org/10.7554/eLife.26161.009

(viscosity) moduli under differential compressive strain (0.01% and 21.5%) using a parallel plate rheometer (*Figure 1—figure supplement 1H*). We found that the stiffness of lrECM did not significantly change (t-test, p=0.579, 0.699) and remained within the range reported for normal breast tissue (*Chaudhuri et al., 2014*; *Paszek et al., 2005*). Therefore, any responses to transient compression in our system were unlikely to be due to increased stiffness of the lrECM.

## Mechanically reverted colonies grow to form organized structures

Non-malignant acini exhibit polarized structures with E-cadherin-containing cell-cell junctions organized into a circular 'star' pattern (*Fournier et al., 2009*). To determine whether compressed colonies contained properly organized cell-cell junctions, we measured the frequency of formation of E-cadherin star patterns (example in *Figure 1B–iii* at 23% strain). In malignant T4-2 cells, compression enhanced the formation of E-cadherin star patterns (*Figure 2D*; analysis of variance p=$3.58 \times 10^{-4}$), demonstrating that compressed colonies exhibited morphologically 'normal' multicellular organization. Quantification of lumen formation (example in *Figure 2A* at 23% strain), another feature of non-malignant acini, confirmed this result (*Figure 2E*; analysis of variance, p=0.0002).

## Mechanical reversion proceeds via nitric oxide signaling

We next investigated molecular mechanisms that could be responsible for mechanical reversion. As previously noted, inhibitory molecules and antibodies against certain oncogenic signaling pathways could induce phenotypic reversion of malignant cells (*Bissell and Hines, 2011*; *Liu et al., 2004*; *Wang et al., 1998*; *Weaver et al., 1997*), and mechanical compression could influence steps involved in these pathways. Recent work has shown that the addition of exogenous nitric oxide (NO) to malignant T4-2 cells reverts the malignant phenotype (*Furuta et al., 2018*). In contrast, S1 cells were found to produce endogenous NO in lrECM culture, and blocking NO production in non-malignant S1 cells could induce malignant behavior. Since mechanical perturbations of other cell types in laminin-containing matrices are known to stimulate NO production (*Gloe and Pohl, 2002*; *Gloe et al., 1999*; *Rialas et al., 2000*), we hypothesized that NO signaling may be the mediator of mechanical reversion. To test this hypothesis, we set out to answer two questions: (1) Does compression stimulate NO production in T4-2 cells? (2) Does inhibiting NO production block mechanical reversion?

To test if compression stimulates NO production in T4-2 cells, we imaged T4-2 cells labeled with the NO sensor dye DAF-FM DA after compression in the 3D-on-top culture geometry (*Figure 3A*). Cells were incubated in media containing 20 µM DAF-FM DA, which reacts with NO to become fluorescent (*Kojima et al., 1999*), for 1 hr before compression (*Figure 3B*). We found that DAF-FM DA fluorescence intensity increased nearly twofold 30–60 min after compression in compressed T4-2 cells relative to uncompressed cells (*Figure 3C*; paired t-test, p<0.03), indicating that compression induced intracellular NO production in malignant cells. This difference was transient; the difference in the NO level was not detectable at 2 hr or 24 hr after compression.

To determine if NO production is necessary for compression-mediated reversion, we treated T4-2 cells with a competitive inhibitor of nitric oxide synthase, L-NAME (*Rees et al., 1990*), applied compression, and cultured cells for 5 days in the 3D-on-top geometry (*Figure 3D*). Measuring colony

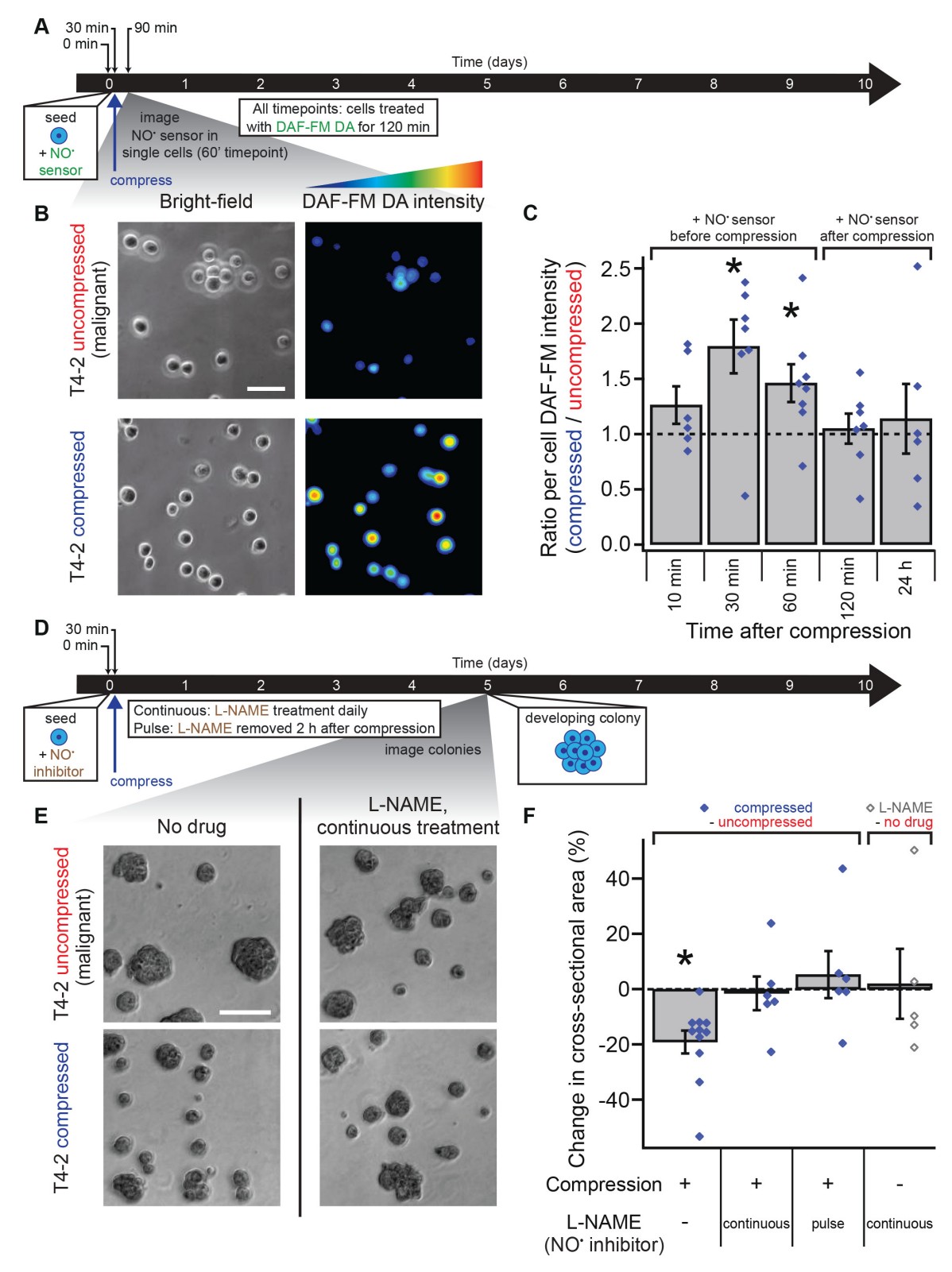

**Figure 3.** Mechanical reversion occurs via nitric oxide signaling. (**A**) Malignant T4-2 cells were compressed 30 min after seeding and coating with 5% lrECM drip in the presence of the nitric oxide (NO) sensor dye DAF-FM DA. At the indicated times after compression, cells were washed with sensor-free media and imaged in wide field fluorescence. For all time points, cells were treated with DAF-FM DA for 2 hr before being washed and imaged. For 10, 30, and 60 min time points treatment with DAF-FM DA began before compression. For 120 min time point, treatment began immediately after

*Figure 3 continued*

compression. For 24 hr time point, treatment did not begin until 22 hr after compression. (B) Bright-field and fluorescence images of compressed and uncompressed malignant T4-2 cells. Scale bar 50 μm. The fluorescent images were pseudocolored from the raw images in ImageJ using one of the default colorscales (blue-to-red). (C) Compression induced NO production in T4-2 cells, as measured by DAF-FM DA fluorescence (N = 6, 7, 8, 7, 6 gels, each with paired uncompressed control, mean ± SEM, with 21 ± 9 cells per sample [mean ± SD]). *Difference is statistically significant at p≤0.03 compared to control samples in paired t-test. (D) T4-2 cells were compressed 30 min after seeding and coating with 5% lrECM in the presence of the NO inhibitor L-NAME. The media was replaced with L-NAME free media after two hours (pulse treatment) or with fresh media containing L-NAME daily (continuous treatment). Five days after compression, cells were imaged. Scale bar 200 μm. (D) Bright-field images of T4-2 colonies, compressed at the single-cell stage and treated continuously with the 0.5 mM L-NAME. Scale bar 200 μm. (F) Treatment of T4-2 cells with the NO inhibitor L-NAME blocked growth sensitivity to compression while L-NAME treatment alone did not alter colony size (N = 11, 6, 6, and 5 gels, points are means of individual experiments, with 176 ± 30 colonies per sample [mean ± SD]). *Difference is statistically significant at p<0.02 compared to uncompressed control samples in paired t-test with a gel made from the same preparation. Treatment of T4-2 cells with NO donor molecules in the absence of compression also yielded smaller colonies (*Figure 3—figure supplement 1*).

DOI: https://doi.org/10.7554/eLife.26161.010

The following source data and figure supplement are available for figure 3:

**Source data 1.** Nitric oxide sensor in T4-2 cells and nitric oxide synthase inhibitor-treated T4-2 colonites.

DOI: https://doi.org/10.7554/eLife.26161.012

**Figure supplement 1.** Pairwise comparison of colony sizes while manipulating nitric oxide conditions and mechanical stimulation.

DOI: https://doi.org/10.7554/eLife.26161.011

sizes using bright-field images (*Figure 3E*) revealed that the phenotypic shift observed in mechanically reverted acini was absent when NO production was inhibited with L-NAME (*Figure 3F* and *Figure 3—figure supplement 1*; paired t-test, p=0.01 and 0.97, respectively). Importantly, treatment of T4-2 cells with L-NAME for only 2 hr after compression was sufficient to abolish reduction in colony size (*Figure 3F* and *Figure 3—figure supplement 1*; paired t-test, p=0.76). The addition of L-NAME to uncompressed T4-2 cells did not alter colony size (*Figure 3F* and *Figure 3—figure supplement 1*; paired t-test, p=0.82), whereas addition of an NO donor SNAP to uncompressed T4-2 cells reduced colony size (*Figure 3—figure supplement 1*; paired t-test, p=0.05). Furthermore, replacing the medium with fresh medium without inhibitor 2 hr after seeding and lrECM polymerization (no compression) produced no measurable difference in colony size (*Figure 3—figure supplement 1*; paired t-test, p=0.39). Finally, supplementing only the starting medium with the NO donor spermine NONOate, which has a half-life of 40 min (*Ramamurthi and Lewis, 1997*), also reduced colony size in uncompressed T4-2 cells (*Figure 3—figure supplement 1*; paired t-test, p=0.02). These results altogether suggest that compression induces a burst of NO production in malignant cells that promotes a phenotypically normal developmental program. This burst in NO production likely activates the same developmental program as the one induced by addition of exogenous NO to malignant cells and the one blocked by inhibition of NO production in non-malignant cells (*Furuta et al., 2018*). In contrast to chemical reversion, which often requires continuous drug treatment throughout the culturing period, NO-mediated mechanical reversion requires only a transient signal to effect long-timescale phenotypic changes.

## Transient compression restores normal developmental behaviors in malignant cells

We next investigated whether transient compression influenced the dynamics of acini development. Non-malignant cells in lrECM rotate coherently in small clusters during the coordinated assembly of an organized, polarized acinus, but malignant cells do not, except when treated with chemical reverting agents (*Tanner et al., 2012*). Because such coherent axial rotation is essential for breast cells' ability to form acinar structure, we hypothesized that it will be restored by compression in malignant cells. Using time-lapse microscopy (*Figure 4A*), we observed that compression indeed restored coherent rotation after the first mitosis in T4-2 cells (*Figure 4B*, paired t-test p=0.009, *Figure 4—videos 1* and *2*). We did not observe any clear trends in the orientation of cell division or direction of coherent rotation that correlated with the direction of compression. Since non-malignant cells undergo first mitosis later than malignant T4-2 cells (*Tanner et al., 2012*) and sustained compression has been shown to slow proliferation rate (*Cheng et al., 2009*), we explored whether compression altered the growth rate of malignant cells. Time-lapse microscopy revealed that

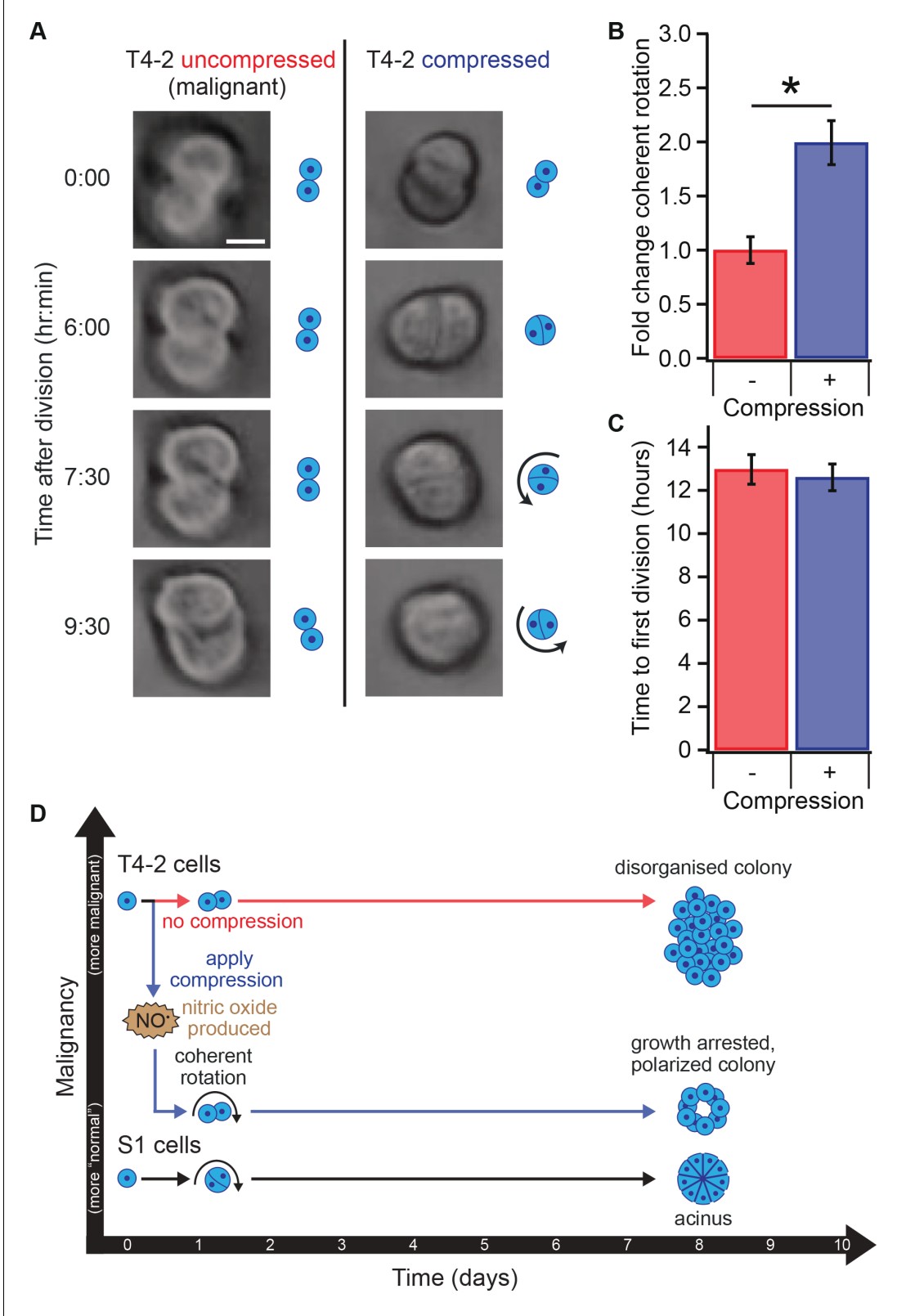

**Figure 4.** Compression allows restoration of coherent rotation after division. (**A**) Time series of malignant T4-2 cells growing in uncompressed and compressed gels. Scale bar 10 μm. Time series were taken from *Figure 4—Video 1* and *Figure 4—Video 1*. (**B**) Cells at the two-cell stage rotated coherently more often in compressed matrices. Fold change coherent rotation is the ratio of the fraction of cell doublets that exhibited rotation with compression to the fraction that exhibited rotation without compression (Fold change = 1, for uncompressed gels). *Fold change is statistically

*Figure 4 continued on next page*

*Figure 4 continued*

significant with p<0.01 in a paired t-test with control samples made from the same preparation and imaged simultaneously. Coherent rotation behavior depends on proper cell-cell contacts, and treatment of cells with function-blocking E-cadherin antibodies disrupted sensitivity to compression (*Figure 4—figure supplement 1*). (C) Time to first division of T4-2 cells was not sensitive to compression. For **B** and **C**, N = 4 gels, each with paired control, mean ± SEM, with 49 ± 29 cell doublets per sample (mean ± SD). *Fraction of cell doublets that underwent coherent rotation is statistically significant with p<0.02 in a paired t-test with control samples made from the same preparation and imaged simultaneously. (**D**) Compression induces nitric oxide production in single breast epithelial cells, which activates 'normal' signaling and development and diminishes the malignant character of colonies days after the compression occurred.

DOI: https://doi.org/10.7554/eLife.26161.013

The following video, source data, and figure supplement are available for figure 4:

**Source data 1.**

DOI: https://doi.org/10.7554/eLife.26161.015

**Figure supplement 1.** E-cadherin function is required for mechanical reversion.

DOI: https://doi.org/10.7554/eLife.26161.014

**Figure 4—video 1.** Uncompressed malignant T4-2 doublets undergo incoherent movement.

DOI: https://doi.org/10.7554/eLife.26161.016

**Figure 4—video 2.** Compression restores coherent rotation of malignant T4-2 doublets.

DOI: https://doi.org/10.7554/eLife.26161.017

---

compression did not detectably delay the time of first mitosis in T4-2 cells (*Figure 4C*; paired t-test p=0.68). Thus, the compression response is not likely the result of direct inhibition of proliferation. Blocking the formation of E-cadherin cell-cell junctions blocked mechanical reversion (*Figure 4—figure supplement 1*). Furthermore, applying compression 24 hr after seeding elicited no change in colony size (*Figure 3—figure supplement 1*).

## Discussion

Our results indicate that mechanical reversion is developmentally related to normal acinar morphogenesis and chemical reversion, in that it requires formation of proper cell-ECM contacts leading to coherent rotation at the two-cell stage early in development of the tissue (*Bissell and Hines, 2011*; *Fournier et al., 2009*; *Liu et al., 2004*; *Tanner et al., 2012*; *Wang et al., 1998*; *Weaver et al., 1997*). Importantly, we observed mechanical reversion only when compression was applied shortly after seeding, and no response was observed to compression applied 24 hr after seeding. This finding is consistent with the existence of a narrow window of time during which epithelial cell polarity is established once cells undergo their first division (*Tanner et al., 2012*; *Ragkousi and Gibson, 2014*). This reversion was heterogeneous, possibly due to the inherent heterogeneity of tumor cells and lrECM, as well as the inhomogeneous nature of transient strain in our system. Since the compression force rapidly dissipated (minutes timescale) and ECM stiffness remained unchanged during compression, we believe transient compression triggered NO-mediated signaling that led to a sustained biochemical change (*Figure 4D*).

Recent work in breast epithelial cells has shown that NO signaling sits at the intersection of the feedback loop between laminin in the ECM and p53, which is disrupted in malignant tissues (*Furuta et al., 2018*). In other cell types, NO has been shown to inhibit malignant behavior via down-regulation of NFκB-mediated MMP-9 expression (*Knipp et al., 2004*; *Sinha et al., 2006*; *Upchurch et al., 2001*), inhibition of calcium-induced Ras/c-RAF/ERK1/2 pathway (*Raines et al., 2004*), and regulation of p53 activity (*Forrester et al., 1996*; *Hofseth et al., 2003*; *Lala and Chakraborty, 2001*; *Wang et al., 2002*). These findings suggest that NO at the appropriate intracellular concentration can serve as a 'native' reverting agent, and that mechanical stimulus by means of compression is one way to activate it. We anticipate that other types of mechanical strains that stimulate nitric oxide production (e.g. shear, stretch and etc.) may also yield phenotypic reversion of malignant cells.

Roles for regulated NO signaling in normal growth and morphogenesis have also been reported in neurons (*Nikonenko et al., 2013*), gametes (*Gelaude et al., 2015*; *Jeseta et al., 2012*), and vascular endothelial cells (*Ramasamy et al., 2015*). In these examples in healthy cell types, a transient NO signal precedes a persistent asymmetry in cell morphology. In tumor cells, the roles of NO

signaling vary widely with concentration, ranging from increasing proliferation and invasiveness to activating apoptosis and inducing resistance to therapeutics (*Dave et al., 2014*; *Fukumura et al., 2006*; *Krasnapolski et al., 2015*; *Kumar et al., 2016*; *Ranganathan et al., 2016*; *Singh and Gupta, 2011*; *Thomas et al., 2008*). Precise measurements of intracellular NO production and tightly controlled dosing with NO donor molecules will allow us to untangle these diverse downstream functions in future studies.

With a role for NO signaling in ECM-to-nucleus communication, we hypothesize that a cell-ECM receptor is the key mechanosensory element in mechanical reversion. NO synthase 1 (NOS-1) binds to the cell membrane and forms a complex with the laminin receptor dystroglycan, the adaptor protein dystrophin and the actin cytoskeleton; this complex is essential for NO production by NOS-1 (*Rando, 2001*). Although T4-2 cells express the proteins in the complex, NO production is impaired (*Muschler et al., 2002*), *Furuta et al., 2018*). In T4-2 cells, the glycosylation of the extracellular domain of dystroglycan is aberrant. We hypothesize that transient compression promotes engagement of laminin and dystroglycan despite aberrant glycosylation. In this way, the cell-ECM receptor dystroglycan itself would serve as a mechanosensor that drives downstream polarity. A conceptually similar mechanism has been observed in C. elegans where hemidesmosomes serve as mechanosensors that promote polarized epithelial morphogenesis (*Zhang et al., 2011*).

It is possible that there are other mechanisms by which compression promotes the development of 'normal,' polarized acini from malignant cells. Compressive force could alter secretion and degradation of matrix proteins (*Cox and Erler, 2011*; *Lu et al., 2012*) or directly induce integrin engagement (*Katsumi et al., 2005*). Regulation of cell-cell signaling (*Curry et al., 2005*; *van Es et al., 2005*) and apoptosis (*Cheng et al., 2009*; *Debnath et al., 2002*), however, are not likely to be directly affected by compression in our assay because the timing of the compression, timescales of mechanical dissipation, and nitric oxide decay are much shorter than these processes. Long timescale behaviors are indeed altered by transient compression, so a mechanism by which some record of the compression event is stored must exist. One possibility is that NO alters cell-cell or cell-ECM interactions via direct NO-mediated post-translation modifications such as nitrosylation or directly activates or inactivates an intracellular signaling protein outside of the p53 pathway (*Smith and Marletta, 2012*). Furthermore, it has been previously observed that the invasive behavior of a sheet of epithelial cells increases under constant compression perpendicular to the primary plane of the sheet (*Tse et al., 2012*), highlighting the importance of considering the direction, timescale, and biological context of a mechanical perturbation when observing a cell or tissue's response behavior.

In endothelial cells, nitric oxide production is mechanically activated on millisecond timescales (*Gloe and Pohl, 2002*; *Gloe et al., 1999*), relaxing actomyosin contraction in nearby smooth muscle cells on sub-second timescales (*Lincoln et al., 1994*; *Palmer et al., 1987*). This response proceeds via nitrosylation of the heme group of soluble guanylate cyclase by an NO radical and is rapidly reversible when NO is depleted (*Derbyshire and Marletta, 2012*). Epithelial cells exhibit increased contractility (*Saez et al., 2005*) and malignant behavior (*Paszek et al., 2005*) in stiff microenvironments by activating Rac1 and PI3K signaling (*Chaudhuri et al., 2014*). Interestingly, Rac1 activity is regulated by NO via nitrosylation (*Raines et al., 2007*). We speculate that a critical balance between NO signaling and contractility may be necessary for normal acinar development, and that mechanical reversion restores this balance in malignant cells. This speculation grows more intriguing when considering the protective effect of breastfeeding against cancer (*Veral et al., 2002*) and the short timescale contractions of myoepithelial cells that surround the luminal epithelial cells during milk letdown (*Adriance et al., 2005*). Could a mechanically activated NO signal from the luminal epithelial cells serve as an 'off' signal for contractility of the myoepithelial cells during milk letdown? Could this mechanical activity be contributing to the protective effect of breastfeeding? Future exploration of the broad parameter space of timescales, magnitudes, and geometries for mechanical stimuli are needed to pursue these intriguing questions.

In summary, we find that a transient compression applied to single malignant cells embedded in a laminin-rich gel can lead to phenotypic reversion of malignant breast cells. Compressed malignant cells show an increase in nitric oxide production relative to uncompressed malignant cells, and blocking NO production also blocks mechanical reversion. Compressed malignant cells undergo coherent rotation at the two-cell stage and require E-cadherin function to achieve mechanical reversion. We propose that the transient forces are sensed through activation of NO signaling in the single cell, allowing activation of p53 and engagement of the normal morphogenetic program. Detailed study

of how NO is produced in response to compressive strain and how it feeds into downstream signaling pathways will help uncover how transient mechanical cues can translate into long-term phenotypic change.

## Materials and methods

### Cell culture

Cell lines of the HMT3522 breast cancer progression series were provided by O.W. Petersen (Laboratory of Tumor Endocrinology, The Fibiger Institute, Copenhagen, Denmark) (*Briand et al., 1996*). HMT3522 cell lines were authenticated by genome sequencing by the provider; mycoplasma tests were negative. Cell lines of the MCF10A cancer progression series were obtained from Karmanos Cancer Institute (MI, USA) under Material Transfer Agreement. MCF10A cell lines were authenticated by the provider; mycoplasma tests were negative.

Culture of non-malignant HMT3522-S1 (RRID:CVCL_2499) and malignant HMT3522-T4-2 (RRID: CVCL_2501) mammary epithelial cells was performed as previously described (*Briand et al., 1996*; *Briand et al., 1987*; *Lee et al., 2007*; *Petersen et al., 1992*). Briefly, cells were cultured at 37°C and 5% $CO_2$ on collagen I-coated tissue culture flasks until embedded in laminin-rich extracellular matrix (lrECM) or seeded in 3D-on-top geometry (*Lee et al., 2007*). Cells in 2D and 3D were cultured in a 1:1 mix of Dulbecco's Modified Eagle's Medium and Ham's F-12 (UCSF Cell Culture Facility). Medium contained insulin, transferrin, sodium selenite, β-estradiol, hydrocortisone and prolactin. Media for S1 cells also contained epidermal growth factor. The laminin-rich extracellular matrix (lrECM) was Matrigel lots A7750, 04147, 36819, and 36147 with protein concentration ranging from 9.2 to 9.4 mg/mL (BD Biosciences). Culture of non-malignant MCF10A (RRID:CVCL_0598) and malignant MCF10A-CA1d cells (RRID:CVCL_6679) was performed as previously described (*Debnath et al., 2003*).

Nitric oxide production was inhibited with the addition of 0.5 mM Nω-nitro-L-arginine methyl ester (L-NAME; Sigma N5751) (*Rees et al., 1990*) to media, with replacement of media daily. To determine this working concentration of L-NAME, T4-2 cells in 3D embedded culture were chemically reverted by treatment with 4 μM LY294002 (Calbiochem 440202), a PI3K inhibitor, as previously described (*Liu et al., 2004*). L-NAME was added to the LY294002-containing media at concentrations from 0 to 10 mM. After 10 days in culture, colonies were imaged in bright field. At concentrations of 0.3 to 1 mM L-NAME, chemical reversion was overcome and malignant growth was restored. Fresh dilutions of the nitric oxide donor molecule S-nitroso-N-acetyl-DL-penicillamine (Santa Cruz Biotech, sc-200319) in medium to a final concentration of 10 μM were applied to cultures daily (*Lander et al., 1993*). Fresh dilution of the nitric oxide donor molecule spermine NONOate (Santa Cruz Biotech, sc-202816) in medium were applied to cultures on day 1 of culture to a final concentration of 5 μM (*Maragos et al., 1991*).

Treatment with E-cadherin function blocking antibody (*Watabe et al., 1994*) was performed as previously described (*Fournier et al., 2009*; *Tanner et al., 2012*). Cells were gently centrifuged and resuspended in a solution of mouse anti-E-cadherin (Thermo Fisher, Rockford, IL 13–1700). This solution was mixed and resuspended in lrECM. The final E-cadherin antibody concentration was 200 μg/mL with 625,000 cells/mL. Control experiments were performed with a mouse IgG control antibody (BD Pharmingen 555749).

### Compression

After embedding cells or seeding and coating, lrECM gels were compressed in custom stretchable wells made of poly-dimethylsiloxane, similar to those described previously (*Brownfield et al., 2013*). Wells were made of Sylgaard 184 (Dow Corning) polymerized at a 9.5:1 ratio of base to curing agent. This mixture was poured into custom-made laser-cut acrylic molds and polymerized at 60°C overnight (laser cutter: ULS2.0 Engraver, clear cast acrylic: McMaster-Carr). Silicone wells were cleaned under ultraviolet light for seven minutes, washed in distilled, deionized water for at least four days under gentle vacuum, and then washed with the cell culture media (without additives) at 37°C for another 2 days or more before use.

Immediately prior to use, wells were stretched using custom-made laser-cut acrylic frames and stainless steel dowel pins (McMaster-Carr). For 3D embedded culture geometry, cells were

resuspended in ~100% lrECM and poured into the stretched well (200 µL) and polymerized at 37°C, 5% carbon dioxide. After 30 min of polymerization, the stainless steel dowel rods were removed with pliers to apply a step compression to the matrix. Media was added, and the wells were returned to the incubator. Media was changed every other day during growth (or every day for L-NAME treatment experiments). Each compressed gel had a matched uncompressed control gel made with the same cell-matrix mixture and the same silicone well preparation. For 3D-on-top culture geometry, cell-free lrECM (100 µL) was poured into the pre-stretched well and polymerized at 37°C, 5% carbon dioxide for 30 min, cells were plated on the lrECM and allowed to attach for 30 min, and then coated with a layer of 5% lrECM in media. The 5% lrECM top layer was left undisturbed for 30 min at 37°C, 5% carbon dioxide to allow polymerization of an insoluble layer of lrECM on top of the cells. After 30 min, compression was applied and media added as for the 3D-embedded cultures.

Applied strain was varied by changing the amount of initial stretch applied to the well before compression by varying the dimensions of the custom acrylic frames. Strain ranged from 0% to 23% compression as measured by photographs of the stretched wells along the primary axis of stretch (*Figure 1—figure supplement 1F*).

## Bright-field microscopy for colony size measurements

After 5 or 10 days of growth, colonies were imaged directly in the wells with a 10x objective on a Zeiss Axiovert 200. Fifty fields of view were taken for each gel to measure colony size. Information identifying samples was removed from the images, and blinded observers measured colony size by manually tracing projected area through a central section of a colony in ImageJ (National Institutes for Health; RRID:SCR_003070). For each gel, an average colony size was measured and compared to the matched control. For initial compression experiments, statistical significance was measured by one-way analysis of variance with Tukey-Kramer multiple comparison test against the null hypothesis that the mean areas are equal. Due to variability in mean colony area measurements from preparation-to-preparation (for uncompressed T4-2 cells, colony area $9200 \pm 3200$ µm$^2$, mean $\pm$ SD, 25 gels), we compared mean colony size for compressed gels directly to their matched uncompressed controls when possible, testing against the null hypothesis that the difference between members of each pair is equal to zero. For dose-dependent experiments, statistical significance at each strain dosage was measured by paired t-test between compressed samples and matched controls. For E-cadherin function blocking experiments, statistical significance under each antibody condition was measured by paired t-test between compressed samples and matched controls.

## Immunostaining

Colonies were fixed and stained as previously described (*Lee et al., 2007*) after 10 days of growth. Media was aspirated and the cell-matrix mixture was directly smeared onto glass microscope slides. These smears were partially dried at room temperature, but not completely dried out (~20 min). Smears were then fixed in 4% paraformaldehyde in phosphate-buffered saline for 30 min and washed with phosphate-buffered saline for 10 min. Cell membranes were then permeabilized in 0.5% Triton X-100 (in distilled, deionized water) for 30 min before 1 hr blocking with 3% bovine serum albumin (in phosphate-buffered saline). Primary antibodies were incubated overnight at 4C in the same blocking solution. Cultures were then washed with phosphate-buffered saline two times (10 min/wash). Hoechst and secondary antibodies were incubated in blocking solution for 2 hr, and cultures were washed with phosphate buffered saline three more times. The samples were mounted on cover slips with ProLong Gold Anti-fade reagent (Invitrogen).

Primary antibodies used were mouse anti-human E-cadherin (1:500, BD Biosciences 610182; RRID:AB_397581) and rabbit anti-human Ki67 (1:400, Vector VP-K451; RRID:AB_2314701). Secondary antibodies used were Alexa 488 and Alexa 568 anti-mouse, anti-rat, and anti-rabbit (all 1:250, Invitrogen). DNA was stained with Hoechst (1:5000) during secondary antibody incubation.

## Immunofluorescence imaging for cell counts, fluorescent labels

Confocal immunofluorescence images were taken with a 20x oil-immersion objective on a Zeiss 710 laser-scanning confocal microscope. Multichannel image stacks were acquired for each colony labeled nuclei and fluorescent proteins of interest. Cell count was measured by counting labeled nuclei from a stack of confocal images of each complete acinus or colony using thresholding and

watershedding methods in ImageJ (modified from count_3D_nuclei_v2.txt by Vytas Bindokas, Univ. of Chicago); the algorithm merged overlapping nuclei in consecutive images within the stack. Presence of Ki67, lumen, and E-cadherin structure was measured manually in blinded fashion. Information identifying samples was removed from the images, and blinded observers scored colonies as:

1. Positive for Ki67 immunostaining or not (example in *Figure 1B–ii*).
2. Containing a cleared lumen or not (example in *Figure 2A* at 23% strain).
3. Having a 'star pattern' in the E-cadherin stain through the central section of the colony or not (example in *Figure 1B–iii*).

These counts were used to calculate the fraction of total colonies of a given sample condition having these features. Statistical significance was determined by paired t-test, between compressed samples and matched control.

## Nitric oxide sensor imaging and analysis

After trypsin treatment prior to plating in 3D-on-top culture geometry, cells were washed and resuspended in media containing 20 µM 4-amino-5-methylamino-2′,7′-difluorofluorescein diacetate (DAF-FM DA; Invitrogen D23844) 27. DAF-FM DA was maintained at this concentration through plating, application of lrECM drip, and compression (1 hr total) before cultures were washed briefly with DAF-FM DA-free media, and imaged in bright-field and widefield fluorescence with a 20x objective on a Zeiss Axiovert 200 at the indicated time points. For 24 hr time points, DAF-FM was not applied until the following day, 1 hr before imaging. In ImageJ, each individual cell (with area $a$) and a local background region were traced in bright-field images and total cell intensity ($I_0$) and mean pixel background intensity ($b$), were measured in the corresponding fluorescence image. Background-subtracted intensity for each cell ($I$) was calculated as $I = I_0 / (ab)$. Normalized intensity for each cell ($I_N$) was calculated as $I_N = I / <I_U>$, where $<I_U>$ was the mean background-subtracted intensity for uncompressed cells in a paired gel experiment.

## Time-lapse microscopy and analysis

Time-lapse microscopy was performed in a custom-built microscope inside a cell culture incubator. This microscope used an electrically shuttered green LED (Phillips Luxeon Rebel), a CMOS camera (DCC1545M, Thorlabs), and a $10 \times 0.25$ NA objective (Nikon) to perform bright-field microscopy. An encoded XY stage and a motorized z-focusing mechanism (Prior Scientific) were used to take measurements at multiple positions simultaneously. After compression, gels were placed in a custom-made 3D-printed ABS plastic holder and put into the time-lapse microscope. The system took approximately 1 hr to equilibrate, and then images were taken at every 10 min. Time-lapse microscopy was stopped after 50 hr. Blinded observers measured the time to first cell division and rotation direction of single cells and doublets. In each separate experiment, at least five fields of view and a minimum of 50 cells in total were measured for each condition. Statistical significance was determined by paired t-test, between compressed samples and matched controls.

## Mechanical testing

Stress relaxation tests were performed on an Electroforce 3200 (Bose) using a 50 g load cell (Honeywell Sensotec) and custom made 1' cylindrical aluminum compression platens. The lower compression platen was pre-heated to 37°C using feedback-controlled thermistors and resistive heating elements (Warner Instruments TC-324B, 64–0106, 64–0274 RH-2). The distance between the upper and lower compression platen was calibrated after pre-heating for 30 min. A droplet of lrECM (100 µL) was placed on the pre-heated lower platen, and the upper platen was immediately brought down to contact the lrECM droplet. Space between the platens was held at 0.4 mm, and the gel was allowed to polymerize for 30 min. This led to formation of a 0.4 mm tall gel with cross-sectional area of 250 mm$^2$. Compression was applied at a rate of 0.05 mm/s for deformation of 0.04 mm (10% strain). Strain rates were chosen to approximately mimic strain rates in the stretchable wells (10%–20% s-1). Load was measured for 40 min, by which time a residual load could not be measured. Relaxation time constants were measured by measuring the amount of time to reach five time constants worth of decay from peak stress (99.4% decay). Our measurements showed that the stress generated by the compressive strain relaxes within a few minutes (*Figure 1—figure supplement*

1G), demonstrating the viscoelastic nature of the lrECM gel (*Allen et al., 2011*; *Chaudhuri et al., 2014*) and the transient nature of the applied compression.

In order to evaluate the strain above which the lrECM strain stiffened (*Pryse et al., 2003*), storage and loss moduli were measured by taking shear amplitude sweeps on a parallel plate rheometer (Anton Paar MCR302). The testing environment consisted of a quartz lower plate and an 8 mm diameter stainless steel upper plate. Plates were pre-heated to 37°C and humidified using a water jacket-heated environmental chamber. lrECM was polymerized in similar fashion to stress relaxation tests, except that gels were 0.4 mm tall and 200 mm$^2$. Storage and loss moduli were measured from 0.01% to 1000% shear strain. This strain regime was chosen to ensure that material breakdown occurred and was measurable. Within this strain regime, we measured an elastic modulus (~200–300 Pa), which compares well with previously reported values for lrECM (~600 Pa) obtained using both a novel, local interferometry technique (μm scale) and bulk rheology measurements, though considerable heterogeneity was found in local measurements (*Reed et al., 2009*). Moduli at 0.01% and 21.5% strain were compared using a two-sided t-test to determine if material properties changed in the regime of interest. We found no significant strain stiffening (two-sided t-test, p=0.579, 0.699), consistent with previously reported mechanical behavior of lrECM gels (*Allen et al., 2011*).

## Acknowledgements

We would like to thank the members of the Fletcher and Bissell Laboratories for their helpful comments and advice, especially WP Ng, KM Chan, MD Vahey, and A Lo. We thank Professor Sanjay Kumar's laboratory for use of their rheometer and Professor Michael Marletta for discussions on nitric oxide. The Bose Electroforce machine was part of the California Institute for Regenerative Medicine Shared Laboratory at UC Berkeley. This work was funded by fellowship from NIH/NIGMS (F32 GM101911, BLR) and by grants from NIH/NCI (PS-OC 60467763–112063-E, MJB and DAF) and NSF (CMMI-1235569, DAF).

# Additional information

### Funding

| Funder | Grant reference number | Author |
|---|---|---|
| National Institute of General Medical Sciences | F32 GM101911 | Benjamin L Ricca |
| National Science Foundation | CMMI-1235569 | Daniel A Fletcher |
| National Cancer Institute | PS-OC 60467763-112063-E | Mina J Bissell<br>Daniel A Fletcher |

The funders had no role in study design, data collection and interpretation, or the decision to submit the work for publication.

### Author contributions

Benjamin L Ricca, Gautham Venugopalan, Kandice Tanner, contributed to original ideas, designed experiments, performed experiments, analyzed data, and wrote the manuscript; Saori Furuta, contributed to original ideas, design of experiments, performing experiments, and writing of the manuscript; Walter A Orellana, Clay D Reber, Douglas G Brownfield, contributed to performing experiments, analysis of data, and writing of the manuscript; Mina J Bissell, Daniel A Fletcher, contributed to original ideas, designed experiments, and wrote the manuscript

### Author ORCIDs

Benjamin L Ricca (iD) https://orcid.org/0000-0003-2958-0662
Saori Furuta (iD) http://orcid.org/0000-0003-1121-0487
Mina J Bissell (iD) https://orcid.org/0000-0001-5841-4423
Daniel A Fletcher (iD) http://orcid.org/0000-0002-1890-5364

**Decision letter and Author response**
Decision letter https://doi.org/10.7554/eLife.26161.020
Author response https://doi.org/10.7554/eLife.26161.021

## Additional files

**Supplementary files**
• Transparent reporting form
DOI: https://doi.org/10.7554/eLife.26161.018

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
