## [Decision Letter]

Thank you for submitting your article "Transient external force induces phenotypic reversion of malignant epithelial structures via nitric oxide signaling" for consideration by *eLife*. Your article has been reviewed by two peer reviewers, and the evaluation has been overseen by a Reviewing Editor and Fiona Watt as the Senior Editor. The reviewers have opted to remain anonymous.

The reviewers have discussed the reviews with one another and the Reviewing Editor has drafted this consolidated review to help define the key issues we believe must be addressed.

Summary:

The authors demonstrate that a transient application (timescale of 10s seconds) of mechanical compression to single mammary epithelial cells significantly decreases their propensity to display malignant growth phenotypes in organoid cultures (timescale of days), and instead promotes the formation of growth-arrested acini. Compression activates NO signaling, and transient NO signaling is required and sufficient for growth arrest and malignant reversion. These observations are intriguing and potentially of significant interest, however a number of questions remain regarding the underlying mechanisms. The major concerns of the reviewers were: 1) the lack of identity of the upstream input/s (sensor) to the NO pathway that are deficient in T4-2 cells in Matrigel-only, but can be mechanically activated and the 2) lack of insights into the mechanism of transient NO signaling that leads to long-term phenotypic reversion.

Essential revisions:

1) Since S1 cells exhibit transient NO signaling during Matrigel culture initiation, however this is lacking in T4-2 cells, this suggests that either conserved environmental sensors (e.g. cell adhesion molecules, cytoskeletal structures) are lacking in T4-2 cells, or their activation is below a threshold, which compression is required to reach. Identifying potential mechanosensors would give insight into the mechanism of the compression response. In this regard, the attempts to uncover mechanism (e.g. blocking E-cadherin) are insufficient, and further are unlikely to be specific to compression-induced malignant reversion, since E-cadherin blockade induced malignant behavior in non-malignant cells.

2) The most intriguing aspect of the work, but also the most lacking in investigation, is how a mechanical signal on the 'seconds' timescale is remembered over the 'days' timescale. miRNAs induced by LM-1 in S1 cells discovered in Furuta et al. should be explored – are these miRNAs induced after compression? If so, what is the kinetics? How long do they act on their target genes?

3) A number of questions arise from their protocol of force application to cells:a) Why was an axial anisotropic stress chosen for the mechanical stimulus? Does the same phenomena occur in the case of equibiaxial (radially-symmetric) compression? Does tensile stress (isotropic or anisotropic) yield similar results with regard to malignant conversion?

b) Force is applied in compression to cell constructs, however measurements of material modulus are done in shear. Do the authors have data (or can point to existing literature) to demonstrate that the modulus measured in shear (under equivalent strains applied in the study) is similar to the modulus measured in tension/compression, which is the nature of their experimental setup?

c) One major puzzle is how the cell response could occur when they are plated "3D-on-top", on top of solid Matrigel in medium containing only 5% Matrigel – surely under these circumstances the cells would not experience any compression at all. A 5% Matrigel solution is liquid and would not transmit a compressive force. Attachment to the solid layer beneath was only for 30 minutes, unlikely to provide meaningful connections, especially as the cells do not spread. Yet the observed phenotype is the same as for embedded cells. This suggests that perhaps there is another event causing the change, not just compression.

4) What are potential physiological origins of a step-application (timescale of seconds) of ~15% compressive strain to cells within tissue? Does this exist? A threshold strain is required to induce the phenotype; does this threshold inform potential underlying molecular mechanisms? Is the magnitude and rate of myoepithelial cell contractions during lactation consistent with their experimental model and results? Significantly more detail, either via literature or through vivo characterization, is needed to support a biological significance.

5) The finding that compression induced coherent rotation in T4-2 cells, similar to S1 cells without compression, is intriguing. What is the kinetics of compression-induced coherent rotation? Does compression directly (mechanically) induce coherent rotation? Does rotation direction and magnitude depend on force orientation, magnitude, rate, etc.? These parameters should be quantified.

6) One puzzle is how the effect could be maintained through multiple cell divisions. I think that the authors need to ask if it continues after acini are retrieved from the Matrigel (e.g. from the 3D-on-top cultures, trypsinized into single cells, and re-embedded. Would this treatment re-establish "malignancy"?

---

## [Author Response]

Essential revisions:1) Since S1 cells exhibit transient NO signaling during Matrigel culture initiation, however this is lacking in T4-2 cells, this suggests that either conserved environmental sensors (e.g. cell adhesion molecules, cytoskeletal structures) are lacking in T4-2 cells, or their activation is below a threshold, which compression is required to reach. Identifying potential mechanosensors would give insight into the mechanism of the compression response. In this regard, the attempts to uncover mechanism (e.g. blocking E-cadherin) are insufficient, and further are unlikely to be specific to compression-induced malignant reversion, since E-cadherin blockade induced malignant behavior in non-malignant cells.

We agree with the reviewer that the full mechanism has yet to be demonstrated, but a strong hypothesis can be formulated based on our results together with the work of Furuta et al. (co-submitted manuscript). Our results demonstrate that a transient compression induces NO production in tumorigenic T4-2 cells and promotes formation of organized structures with apicobasal polarity, which is otherwise defective in these cells.

We have revised our manuscript to clearly state the hypothesis that the mechanosensory element in the fascinating behavior of mechanical reversion is the interaction between dystroglycan and laminin, framing the discussion of our findings in the context of the related literature, with the addition of the following paragraph:

“With a role for NO signaling in ECM-to-nucleus communication, we hypothesize that a cell-ECM receptor is the key mechanosensory element in mechanical reversion. […] A conceptually similar mechanism has been observed in *C. elegans* where hemidesmosomes serve as mechanosensors that promote polarized epithelial morphogenesis (Zhang et al., 2011).”

2) The most intriguing aspect of the work, but also the most lacking in investigation, is how a mechanical signal on the 'seconds' timescale is remembered over the 'days' timescale.

Our work and the work of others have demonstrated that reversion in this breast epithelium model system is only possible in a narrow time window, during the first ~24–36 hours after cells receive initial extracellular cues (Bissell and Hines 2011; Fournier et al., 2009; Liu et al., 2004; Tanner et al., 2012; Wang et al., 1998; Weaver et al., 1997). Consistent with these results, we did not observe mechanical reversion when transient compression was applied 24 hours after seeding T4-2 cells (Figure 3—figure supplement 1).

This critical window exists because epithelial cell polarity is established (or not established) once cells undergo their first division (Tanner et al., 2012; Ragkousi et al., 2014). Hence, we hypothesize that in our experiments the mechanical signal is ‘remembered’ over days because the mechanical stimulus promotes a response to laminin and subsequent coherent rotation that is essential for organizing the basement membrane that guides development over longer timescales.

Acinus-forming S1 cells produce NO in response to laminin in the ECM; NO levels spike within an hour of exposure to laminin and decline thereafter (Furuta et al., 2018 co-submitted). Consistent with this early signaling event, our experiments show that inhibition of NO production by a 2-hour pulse treatment with L-NAME (NOS inhibitor) completely abrogates acinus formation by T4-2 cells in response to transient compression (Figure 3). Thus, the transient force application acting on a time scale of seconds initiated NO signaling to promote the formation of proper cell-cell junctions; this stimulus was sufficient to irreversibly alter the phenotype of T4-2 cells on time scales of days.

We have revised the Discussion in our manuscript to highlight how the timing of the initial signal (in our experiments, transient compression stimulating NO production) is critical to the development of epithelial polarity. We also explain more clearly how our findings relate to the earlier work of Tanner et al., 2012 showing the importance of coherent rotation at early time points for long-term acinar development.

miRNAs induced by LM-1 in S1 cells discovered in Furuta et al. should be explored – are these miRNAs induced after compression? If so, what is the kinetics? How long do they act on their target genes?

Our work and the co-submitted work of Furuta et al. are distinct but complementary studies in that both converge on the role of NO in the morphogenesis of polarized breast epithelium, hence our decision to co-submit them to *eLife*. Our study discovered that phenotypic reversion could result from transient mechanical stimulus. The work of Furuta et al. focused on elucidating biochemical signaling involved in mammary morphogenesis and tumor reversion. Although it would be interesting to test the involvement of the miRNAs identified by Furuta et al. in our system in a future set of experiments, this is beyond the scope of our current study.

We believe there is great value in publishing the novel finding that mechanical compression can signal through the same pathway – NO – as normal mammary morphogenesis to drive phenotypically normal development. The point of co-submitting this work was to offer a more complete picture of mammary development and tumorigenesis together – biochemical and biophysical – without unnecessary duplication. Future work will address the many points of connection between the studies as we work to map out the complete mechanistic connection between force, signaling, and phenotype.

3) A number of questions arise from their protocol of force application to cells:a) Why was an axial anisotropic stress chosen for the mechanical stimulus? Does the same phenomena occur in the case of equibiaxial (radially-symmetric) compression? Does tensile stress (isotropic or anisotropic) yield similar results with regard to malignant conversion?

Axial strain application was chosen as a starting point due to the effects previously observed in our strain application chambers on branching morphogenesis in mammary organoids (Brownfield et al., 2013). Whether the same phenomena occurs in the case of alternative geometries of strain application is an interesting question that merits exploration in future work. In our experiments, transient compression is applied at the single-cell stage, and we observed no correlation between the direction of compression and the direction of rotation. Therefore, we hypothesize that radially-symmetric compression of the appropriate magnitude would yield similar results in stimulating NO production and promoting normal growth and polarity development.

b) Force is applied in compression to cell constructs, however measurements of material modulus are done in shear. Do the authors have data (or can point to existing literature) to demonstrate that the modulus measured in shear (under equivalent strains applied in the study) is similar to the modulus measured in tension/compression, which is the nature of their experimental setup?

Reed et al. reported a median elastic modulus for lrECM of ~600 Pa when measured either locally using a novel interferometry technique or when measured using bulk rheology measurements, though the authors note there is considerable heterogeneity in local measurements (Reed et al., 2009). We have revised our manuscript to cite these findings, which are on the same order as our measurements, in the description of our experiments.

c) One major puzzle is how the cell response could occur when they are plated "3D-on-top", on top of solid Matrigel in medium containing only 5% Matrigel – surely under these circumstances the cells would not experience any compression at all. A 5% Matrigel solution is liquid and would not transmit a compressive force. Attachment to the solid layer beneath was only for 30 minutes, unlikely to provide meaningful connections, especially as the cells do not spread. Yet the observed phenotype is the same as for embedded cells. This suggests that perhaps there is another event causing the change, not just compression.

We apologize for any confusion arising from our description in the methods. For the description of 3D-on-top culture, we have revised the manuscript to clarify that cells were plated on the lrECM and allowed to settle for 30 minutes. Then, the growth medium containing 5% lrECM was applied on the top of cells and left for 30 min to allow for the insoluble layer of lrECM to precipitate over cells (per methods described in Lee et al., 2007). The phenotype of epithelial cells in 3D-on-top culture is very similar to that in embedded culture. Moreover, the 3D-on-top culture has the advantage over embedded culture in that it requires only 4–6 days to observe differences in the colony phenotypes whereas embedded culture requires 8–10 days. Therefore, we have utilized both on-top and embedded cultures almost interchangeably.

We also found that the phenotypic effect of transient compression was similar between the 3D-on-top and embedded culture geometries. As the reviewer suspects, the force would not be transmitted through the layer of 5% lrECM efficiently. On the other hand, 100% lrECM, on which single cells are sitting, would efficiently transmit the compressive force, and we found this was sufficient to alter the phenotype of malignant cells.

4) What are potential physiological origins of a step-application (timescale of seconds) of ~15% compressive strain to cells within tissue? Does this exist? A threshold strain is required to induce the phenotype; does this threshold inform potential underlying molecular mechanisms? Is the magnitude and rate of myoepithelial cell contractions during lactation consistent with their experimental model and results? Significantly more detail, either via literature or through vivo characterization, is needed to support a biological significance.

Our system was designed to probe acini with a defined mechanical input rather than to mimic a specific physiological behavior or state, allowing us to address the fundamental question of whether direct mechanical compression affects tumor reversion. Further exploration of the physiological role of mechanical reversion will entail designing experiments that more closely mimic the natural environment in which breast epithelium experiences forces within the organism, such as breastfeeding. We certainly do not intend to claim we have discovered a novel mechanism by which breastfeeding protects against cancer. Instead, we simply aimed to point out that our findings point to an intriguing line of inquiry that could connect a well-documented medical phenomenon with our new observations. We have clarified the language in our Discussion to make this clear.

5) The finding that compression induced coherent rotation in T4-2 cells, similar to S1 cells without compression, is intriguing. What is the kinetics of compression-induced coherent rotation? Does compression directly (mechanically) induce coherent rotation? Does rotation direction and magnitude depend on force orientation, magnitude, rate, etc.? These parameters should be quantified.

This is an interesting question, and we are interested in exploring the parameter space of force orientation, magnitude, rate, etc. on coherent rotation in future work. As described in Tanner et al.’s work, coherent axial rotation is a hallmark of mammary epithelial cells undergoing acinar morphogenesis in lrECM (Tanner et al., 2012). The paper demonstrated that coherent rotation depends on the cell’s ability to undergo asymmetric cell division and formation of cell-cell junctions during the development of structures with apicobasal polarities from single cells. Despite the consistent occurrence of coherent rotation, the direction and rate of coherent rotation in the whole population is random (Tanner et al., 2012). The phenomenon of coherent rotation is absent in malignant cells. However, when malignant cells undergo phenotypic reversion, where they restore such morphogenetic program, they resume coherent rotation like normal cells (Tanner et al., 2012). This phenomenon appears to be observed under all reverting conditions, including transient compressive force application (our current work), addition of NO, and chemical inhibition of oncogenic pathways (Figure 3; Tanner et al., 2012; Furuta et al., 2018 co-submitted). Taken together, these observations suggest that coherent rotation is not caused by the actions of certain reverting agents but rather is the consequence of the restoration of morphogenetic program. For the purposes of this manuscript, we have reported the first observation that compression induces coherent rotation in T4-2 cells and the connection with other methods of reversion, leaving an exploration of the compression parameter space for future work.

6) One puzzle is how the effect could be maintained through multiple cell divisions. I think that the authors need to ask if it continues after acini are retrieved from the Matrigel (e.g. from the 3D-on-top cultures, trypsinized into single cells, and re-embedded. Would this treatment re-establish "malignancy"?

As described in our second answer to comment 2, cellular signaling in response to the transient compression takes place within a brief period of time and quickly attenuates. We hypothesize that once the initial scaffold of proper cell-cell and cell-ECM contacts are established, the colony’s phenotype has been determined. Consistent with previous studies, this must occur within the first ~24 to 36 hours. Proper organization is established at or before the first cell division event and reinforced during coherent rotation. The phenotypic effect is irreversible and sustained throughout multiple cell divisions, leading to formation of polarized acini by malignant T4-2 cells (i.e., ‘malignant reversion’). The reverted and quiescent phenotype of malignant cells is sustained when the intact colonies are re-plated into another Matrigel (unpublished data, Bissell Laboratory.). However, tumor reversion depends on the restoration of basement membrane-hemidesmosome complexes around each colony and proper cell-cell junctions (Weaver et al., 1997). Once the basement membrane and cell junctions are disrupted (e.g., by trypsinization), formerly reverted cells again exhibit malignant phenotype (unpublished data, Bissell Laboratory).